# Dynamics of Viral Shedding and Symptoms in Patients with Asymptomatic or Mild COVID-19

**DOI:** 10.3390/v13112133

**Published:** 2021-10-22

**Authors:** Seongman Bae, Ji Yeun Kim, So Yun Lim, Heedo Park, Hye Hee Cha, Ji-Soo Kwon, Mi Hyun Suh, Hyun Jung Lee, Joon Seo Lim, Jiwon Jung, Min Jae Kim, Yong Pil Chong, Sang-Oh Lee, Sang-Ho Choi, Yang Soo Kim, Ho Young Lee, Sohyun Lee, Man-Seong Park, Sung-Han Kim

**Affiliations:** 1Department of Infectious Diseases, Asan Medical Center, University of Ulsan College of Medicine, Seoul 05505, Korea; songman.b@gmail.com (S.B.); aeki22@naver.com (J.Y.K.); soyun89lim@gmail.com (S.Y.L.); heyhe0102@naver.com (H.H.C.); kwonjs92@alumni.kaist.ac.kr (J.-S.K.); happymh1984@hanmail.net (M.H.S.); silverspec@naver.com (H.J.L.); trueblue27@naver.com (J.J.); nahani99@gmail.com (M.J.K.); drchong@amc.seoul.kr (Y.P.C.); soleemd@amc.seoul.kr (S.-O.L.); sangho@amc.seoul.kr (S.-H.C.); yskim@amc.seoul.kr (Y.S.K.); 2BK21 Graduate Program, Department of Biomedical Sciences, Korea University College of Medicine, Seoul 02841, Korea; phd0919@naver.com (H.P.); powerkw1@korea.ac.kr (H.Y.L.); leesh9253@korea.ac.kr (S.L.); 3Department of Microbiology, Institute for Viral Diseases, Biosafety Center, College of Medicine, Korea University, Seoul 02841, Korea; 4Clinical Research Center, Asan Institute for Life Sciences, Asan Medical Center, University of Ulsan College of Medicine, Seoul 05505, Korea; joonseolim@gmail.com

**Keywords:** SARS-CoV-2, presymptomatic, viral shedding, subgenomic RNA, viable culture

## Abstract

We conducted a prospective cohort study at a community facility designated for the isolation of individuals with asymptomatic or mild COVID-19 between 10 January and 22 February 2021 to investigate the relationship of viral shedding with symptom changes of COVID-19. In total, 89 COVID-19 adult patients (12 asymptomatic, 16 presymptomatic, 61 symptomatic) were enrolled. Symptom scores, the genomic RNA and subgenomic RNA of SARS-CoV-2 from saliva samples with a cell culture were measured. Asymptomatic COVID-19 patients had a similar viral load to symptomatic patients during the early course of the disease, but exhibited a rapid decrease in viral load with the loss of infectivity. Subgenomic RNA and viable virus by cell culture in asymptomatic patients were detected only until 3 days after diagnosis, and the positivity of the subgenomic RNA and cell culture in symptomatic patients gradually decreased in both from 40% in the early disease course to 13% at 10 days and 4% at 8 days after the symptom onset, respectively. In conclusion, symptomatic patients have a high infectivity with high symptom scores during the early disease course and gradually lose infectivity depending on the symptom. Conversely, asymptomatic patients exhibit a rapid decrease in viral load with the loss of infectivity, despite a similar viral load during the early disease course.

## 1. Introduction

Asymptomatic or presymptomatic transmission is an important characteristic of SARS-CoV-2 infection [1], which provides a scientific basis for the recommendation of the general public to use face masks regardless of the presence of symptoms and the testing of asymptomatic individuals for SARS-CoV-2 in high-risk settings such as nursing homes or healthcare facilities [2]. A recent study reported that about three-quarters of individuals with SARS-CoV-2 infection, who were asymptomatic at the time of diagnosis, remained persistently asymptomatic, and the remaining one-quarter of those individuals developed symptoms during the disease course [3]. Therefore, asymptomatic individuals with SARS-CoV-2 infection should be divided into those who are persistently asymptomatic and those who undergo a presymptomatic period followed by a symptomatic period. An epidemiologic study suggested that persistently asymptomatic cases were associated with about an 80% lower infectivity than symptomatic cases and presymptomatic cases [4]. However, there are limited data on the comparative viral shedding kinetics of persistently asymptomatic patients to those of symptomatic or presymptomatic patients with the strict classification of asymptomatic patients during the disease course.

Past volunteer studies on the influenza virus revealed that viral shedding kinetics largely overlap those of systemic symptoms; thus, showing that symptom scores may be used as a surrogate marker of infectiousness [5]. However, there are limited data on the relationship of viral shedding with symptom changes during the disease course of COVID-19. Empirical data on the dynamics of viral shedding and symptoms could provide an important basis for the assessment of a treatment response and infection control practice. We, thus, evaluate the daily changes in viral shedding and symptom scores in patients with asymptomatic or mildly symptomatic COVID-19.

## 2. Materials and Methods

In South Korea, adult patients with mild or asymptomatic COVID-19 were isolated in non-hospital community facilities regardless of the accompanying symptoms. All COVID-19 patients admitted to the community facilities were diagnosed by PCR testing for reasons such as COVID-19-related symptoms or contact with other confirmed cases. Between 10 January and 22 February 2021, we recruited adult patients with laboratory-confirmed COVID-19 from a designated non-hospital community facility in Seoul, South Korea, who were willing to record their symptoms in an electronic diary. All the patients were asymptomatic or had only mild symptoms at the time of admission. Patients who agreed with daily saliva sample collection were enrolled for the viral kinetic study (Appendix A). Those who were younger than 18 years of age or who were pregnant were excluded. All admitted patients were monitored for symptoms related to COVID-19. Patients were required to check their vital signs (i.e., body temperature, oxygen saturation, and blood pressure) using portable, automatic devices provided at admission and report them to the medical staff twice a day. 

Following the policy of the health authorities, patients were discharged according to the clinical course if their symptoms improved or no longer worsened after 10 days from the date of diagnosis. Patients with newly developed symptoms, such as shortness of breath, desaturation or events requiring hospital treatment while admitted at a community facility were transferred to a hospital facility. All participants provided written informed consent. This study was approved by the institutional review board of Asan Medical Center (IRB no. 2020-0336).

### 2.1. Symptom Score and Saliva Sample Collection

Scores for each of the 25 COVID-19-related symptoms were submitted by patients and recorded in electronic medical records on a daily basis by medical staff. The severity of each symptom was scored from 0 to 3 points as follows: (i) score of 0 (no symptom), (ii) score of 1 (transient or mild discomfort, no interference with daily activity and no requirement of medical intervention or therapy), (iii) score of 2 (mild-to-moderate limitation in daily activity and symptoms are controlled with medical intervention or therapy) and (iv) score of 3 (substantial limitation in daily activity and symptoms are not well controlled with medical intervention or therapy). The questionnaire sheets for COVID-19-related symptoms are summarized in Appendix A. Patients were classified into three groups: (i) symptomatic patients whose COVID-19-related symptoms developed before or at the time of PCR diagnosis, (ii) presymptomatic patients whose symptoms developed after PCR diagnosis and (iii) asymptomatic patients who did not develop any symptom up to 10 days after PCR diagnosis. 

Self-collected saliva samples were obtained from patients from the day of study enrollment until the day of discharge. Each day, patients collected a 2 mL volume of saliva into an airtight container provided at admission. Patients were asked to avoid food, water, and teeth brushing for at least 30 min prior to sample collection. Saliva samples were picked up within an hour by the medical staff and transported to a designated laboratory where they were aliquoted and stored at −80 °C until use. No preservation or transport medium was used.

### 2.2. Laboratory Procedure

The collected saliva samples were inactivated at 65 °C for 30 min in a special negative pressure laboratory. Genomic viral RNA was extracted from the specimens using a QIAamp viral RNA Mini kit (QIAGEN Inc., Hilden, Germany). To determine the SARS-CoV-2 genomic viral RNA copy number, multiplex real-time RT-PCR assays targeting the S and N genes were developed; the primer and probe sequences and detailed procedures of the multiplex real-time RT-PCR assay are provided in Appendix A). 

SARS-CoV-2 N and S gene subgenomic RNAs were detected by multiplex real-time RT-PCR assays. The shared forward primer was designed in the 5′ leader sequence, and reverse primers and probes were located in the gene sequences coding for proteins N and S (Appendix A). Real-time RT-PCR reactions were performed as described in Appendix A. 

Culture-based isolation of SARS-CoV-2 from respiratory specimens that revealed positive genomic RNA results was performed by a plaque assay in a Biosafety Level 3 laboratory at Korea University College of Medicine, Seoul, South Korea. Vero cells were cultured in 6-well plates with 9 × 10^5^ cells/well about 24 h earlier. Specimens were serially diluted 10-fold using PBS. A total of 200 µL of each diluted sample was inoculated into cells and incubated for 1 h (37 °C, 5% CO_2_) with rocking every 15 min, and overlaid with 2 mL of Dulbecco’s Modified Eagle Medium/Nutrient Mixture F12 (DMEM/F-12) medium containing 0.6% oxoid agar. Viral plaque formation was visualized by crystal violet staining after 72 h of incubation at 37 °C, 5% CO_2_ incubator.

### 2.3. Statistical Analysis

Categorical variables were compared using chi-squared or Fisher’s exact test, and continuous variables were compared with the Mann–Whitney U test or Kruskal–Wallis test, as appropriate. The time-wise differences in the dynamics of viral shedding and symptom score between asymptomatic, presymptomatic and symptomatic patients were compared using generalized estimating equations. We assumed an exchangeable working correlation structure in the generalized estimating equation models. Interactions between time and symptom groups were also evaluated. The days from symptom onset and days from COVID-19 diagnosis were used as the time scale. In the asymptomatic group, only the number of days from the diagnosis was used as the time scale. Viral shedding values with less than the lower limit of quantification (LoQ; 2.6 log copies/mL) of the RT-PCR assay but with positive qualitative results were imputed with half of the lower LoQ. Negative RNA values were imputed with 0 log copies/mL. In addition, we performed survival analysis to estimate the negative conversion rate of PCR using Kaplan–Meier plot and log-rank test. All tests of significance were two-tailed and *p* values < 0.05 were considered significant. Data were analyzed using SPSS Statistics for Windows, version 23.0 (IBM Corp., Armonk, NY, USA) or R version 4.0.4 (R Project for Statistical Computing, Vienna, Austria).

## 3. Results

The study flow chart is shown in Appendix A. In total, 89 patients, including 12 (13%) asymptomatic, 16 (18%) presymptomatic, and 61 (69%) symptomatic patients, were enrolled. The median age was 49 years (IQR, 34–60). Of the patients, 86.5% were admitted to the facility on the day of COVID-19 diagnosis, and the remaining 13.5% were admitted to the facility a day later. Except for the age and days from symptom onset to admission, the three groups did not show significant differences in the baseline characteristics (Table 1).

### 3.1. Dynamics of Symptoms in Patients with Mild COVID-19

The frequencies and durations of symptoms during admission in the symptomatic group and the presymptomatic group are summarized in Appendix A. The most common symptoms were a cough (73%) and sputum (71%), followed by a febrile sense (61%), fatigue (61%) and anorexia (61%). There was no significant difference in the frequency of symptoms between the symptomatic group and the presymptomatic group, except that rhinorrhea was more common in the symptomatic group (59% vs. 25%, *p* = 0.02). No significant difference in the frequency of initial symptoms was observed between the two groups, except that myalgia (41% vs. 13%, *p* = 0.041) and headache (48% vs. 19%, *p* = 0.048) were more common in the symptomatic group (Appendix A). The median duration of COVID-19-related symptoms was significantly longer in the symptomatic group (median days, 12 (IQR, 10–13)) than in the presymptomatic group (10 (7.25–11)); *p* < 0.001).

The dynamics of COVID-19-related symptom scores in 89 patients are described in Figure 1. In both the symptomatic group and the presymptomatic group, symptom scores peaked on the earliest day of the assessment after symptom onset. The mean peak symptom score was significantly higher in the symptomatic group compared with the presymptomatic group (8.12 vs. 4.34; *p* < 0.001). In both groups, symptom scores gradually decreased over time after symptom onset (*p* for time effect = 0.007). Symptom scores were higher in the symptomatic group than in the presymptomatic group at all time points (*p* for group effect = 0.001). There was no statistically significant group-by-time interaction effect (*p* for interaction = 0.20).

### 3.2. Viral Shedding Kinetics

Of the 89 patients, 40 patients who agreed to daily saliva sampling were included in the viral kinetics study (7 (18%) asymptomatic, 11 (28%) presymptomatic, and 22 (55%) symptomatic). A total of 316 samples from 40 patients underwent a genomic RNA assay and their viral shedding kinetics are shown in Figure 2a,b. The initial viral loads (log copies/mL) were not significantly different among the asymptomatic, presymptomatic and symptomatic groups (median, 3.24 (IQR, 1.55–4.05) vs. 4.46 (3.18–5.70), vs. 4.12 (2.97–5.07); *p* = 0.40). The peak viral loads were also not significantly different among the asymptomatic, presymptomatic and symptomatic groups (median 4.30, IQR (3.97–4.66) vs. 4.72 (3.73–6.14) vs. 4.98 (3.64–5.65); *p* = 0.50). In all patients, viral loads decreased according to time (*p* for time effect < 0.001) and the viral loads were significantly different among the three groups over time (*p* for group effect = 0.003). No group-by-time interaction effect was detected (*p* for interaction = 0.65). However, the degree of changes from peak viral loads in the asymptomatic group declined more rapidly compared with those in the symptomatic group, as the median (IQR) duration from symptom onset to negative conversion for genomic RNA detection was 6 (4.5–6) days in the asymptomatic group and 9 (6–10) days in the symptomatic + presymptomatic group (*p* by log-rank test = 0.01) (Appendix A). In addition, Kaplan–Meier curves for negative conversion between the two groups were significantly different when the time scale was changed to the number of days from diagnosis (Appendix A).

The time-wise changes in the symptom score and viral loads in the symptomatic group and the presymptomatic group are shown in Figure 2c,d. Although the overall symptom scores were lower in the presymptomatic group than in the symptomatic group, the symptom score and viral shedding curves showed similar shapes in the two groups. A weak but significant correlation was observed between the viral load and symptom score in the symptomatic group (Spearman’s rho, 0.28, *p* < 0.001), whereas no significant correlation was observed between the viral load and symptom score in the presymptomatic group (Spearman’s rho, 0.14, *p* = 0.18; Appendix A).

### 3.3. Subgenomic RNA Viral Shedding and Isolation of SARS-CoV-2 by Culture

We further analyzed subgenomic RNA from all saliva samples to infer the duration and prevalence of viable virus shedding. As shown in Figure 3, subgenomic RNAs were detected for a considerably longer duration in symptomatic or presymptomatic patients compared with asymptomatic patients. In symptomatic and presymptomatic patients, subgenomic RNA was detected up to day 10 and day 8 after symptom onset, respectively, with the percentage of subgenomic RNA positivity gradually decreasing over time. In asymptomatic patients, however, subgenomic RNA positivity was detected only on the 3rd day after diagnosis and not thereafter.

In addition, we performed cell cultures using 144 saliva samples with positive genomic RNA results. As shown in Figure 4, viable virus by cell culture was also detected for a considerably longer duration in symptomatic or presymptomatic patients compared with asymptomatic patients. In symptomatic and presymptomatic patients, the cell culture revealed positive results up to day 8 and day 4 after symptom onset, respectively, with the percentage of cell culture positivity gradually decreasing over time. In contrast, cell cultures from asymptomatic patients exhibited positive results only on the 3rd day after diagnosis and not thereafter.

## 4. Discussion

In this study, we found that patients who experienced a presymptomatic period had lower symptom scores than patients with symptoms at the time of diagnosis, while the viral loads were comparable between the two groups. In both the symptomatic group and presymptomatic group, the time-wise changes in viral shedding kinetics after symptom onset overlapped the changes in the symptom score. Furthermore, while the persistently asymptomatic patients had a comparable degree of viral shedding in the early period after the PCR-based diagnosis, negative conversion occurred more rapidly than symptomatic or presymptomatic patients. In addition, by analyzing subgenomic RNA and viral culture, we observed that the duration of viable virus shedding of asymptomatic patients was shorter than that of symptomatic or presymptomatic patients. Collectively, our results showed that, considering their prolonged duration of infectiousness, symptomatic patients (with or without a presymptomatic period) may have a higher contribution to the ongoing community spread of COVID-19 than asymptomatic patients.

Recent epidemiologic studies consistently suggested that the secondary attack rate of COVID-19 was lower in contacts of people with asymptomatic patients with SARS-CoV-2 infection than those with symptomatic COVID-19 [4,6,7]. A German study investigating 42 household contacts and 212 other contacts from 46 symptomatic patients and 7 asymptomatic patients with SARS-CoV-2 infection reported that the number of secondary cases was 41 from symptomatic patients and 0 from asymptomatic patients [6]. A Chinese study from Wuhan showed that asymptomatic cases had about an 80% lower infectivity than symptomatic cases [4]. Furthermore, a retrospective study in China showed that the secondary attack rate in 1078 close contacts from 185 asymptomatic patients was 1.1%, which was about 75% lower than the secondary attack rate of 4.1% in 3136 close contacts from 393 symptomatic patients [7]. However, the existing epidemiologic data may be prone to underestimating the secondary attack rate of asymptomatic individuals with SARS-CoV-2 infection because such individuals are usually discovered through contact tracing during outbreak investigations or surveillance testing, and are immediately isolated and have less chance to spread to others. In this context, our current comparative data on viral shedding between asymptomatic and symptomatic patients with SARS-CoV-2 infection provide a more practical insight into the infectivity of asymptomatic individuals with SARS-CoV-2 infection. Our data showed that the duration of genomic, subgenomic and culture-based viral shedding was considerably shorter in asymptomatic patients than in symptomatic patients with or without a presymptomatic period. Therefore, our findings provide crucial support to the recent epidemiologic findings and suggest that the contribution of asymptomatic transmission to the ongoing community spread of COVID-19 is relatively lower than that of symptomatic or presymptomatic transmission.

In influenza studies, clinical symptom scores have been widely used to assess the treatment response in patients and the value of the isolation policy because influenza viral shedding kinetics largely overlap those of systemic symptoms [5]. However, as for COVID-19, limited data are available on the relationship of viral shedding with symptom changes during the disease course. We found that, similar to the changes in the symptom score in influenza patients, the symptom score in symptomatic COVID-19 patients consistently decreased during the disease course (Figure 1). The decline of the symptom score in symptomatic COVID-19 revealed a relatively protracted pattern compared with that in influenza (near 0 one week after the symptom onset in influenza) [5]. Therefore, our finding provides a possibility of using the symptom score system as an objective measure for assessing clinical improvement in future clinical studies. Interestingly, symptomatic COVID-19 patients who experienced a presymptomatic period exhibited significantly lower symptom scores than those without a presymptomatic period, while the viral loads were comparable between the two groups. Further studies are needed to identify the risk factors that may be used to differentiate symptomatic COVID-19 from presymptomatic COVID-19 followed by symptoms.

The culture-based isolation of SARS-CoV-2 is regarded as the most direct way of determining the presence of a replicating virus that has a transmission ability. However, culture-based isolation is difficult, labor-intensive and time-consuming, and has suboptimal sensitivity for detecting viable viruses due to bacterial contamination or cell detachment. Accordingly, previous studies suggested that a subgenomic RNA assay may be useful for predicting the duration of infective viral shedding because this assay may be complementary to suboptimal sensitivity of cell culture for viable virus shedding [8,9,10,11]. Our previous study also found that, while subgenomic RNA was detected for a few days after the negative conversion of viral culture, the mean duration of viral shedding assessed by subgenomic RNA detection was notably similar to that of virus culture [11]. Therefore, the true rate of viable SARS-CoV-2 shedding would lie between the sensitivity of cell culture-positive samples and that of subgenomic RNA-positive samples. We, therefore, evaluated subgenomic RNA viral shedding as a potential surrogate for viable virus shedding in asymptomatic and symptomatic patients with SARS-CoV-2 as well as cell culture, and found that subgenomic RNA and viable virus by cell culture in asymptomatic patients was detected only in the early period after diagnosis, while that in symptomatic patients was detected until 10 days after symptom onset (Figure 3 and Figure 4). Taken together, our results show that asymptomatic individuals with SARS-CoV-2 infection may have infectivity only during a short time period in the early disease course, while symptomatic patients with SARS-CoV-2 infection may have a more durable infectivity before and after symptom onset.

This study had several limitations. First, we could not collect the respiratory samples during the presymptomatic period in symptomatic patients who experienced a presymptomatic phase. All the presymptomatic patients who did not have a symptom at the time of diagnosis and facility admission developed symptoms immediately after the admission. As such, we could not assess how long viral shedding had occurred prior to symptom onset in the presymptomatic group. Human volunteer challenge studies or the collection of respiratory samples from close contact during quarantine may provide an answer to this important question. Second, the absence of symptom scores from diagnosis to facility admission hindered the assessment of the very early symptom score changes in symptomatic patients without a presymptomatic period. However, since the majority of patients were admitted to our facility on the day of the diagnosis, the limited assessment of the very early symptoms may be applied to only a small proportion of patients. Third, although the symptom score was used as a continuous variable in our study, the severity of symptoms could not be scaled arithmetically at equal intervals, and the score for each symptom was not interchangeable. Nevertheless, scoring symptoms hold value as they have been pragmatically used for respiratory viral infections such as rhinovirus and the influenza virus [12,13,14]. Fourth, the secondary attack rate according to the presence of symptoms could not be calculated due to the absence of epidemiologic data. Fifth, we could not evaluate the dynamics of viral shedding and the symptom score of various genetic variants of SARS-CoV-2 such as the delta variant because there were no reported cases of the delta variant and only a limited number of cases infected with variants of concern in South Korea during the study period [15]. In addition, breakthrough SARS-CoV-2 infection in vaccinated individuals could not be evaluated because COVID-19 vaccinations had not begun in South Korea during the study period. Further studies are needed in this area. Sixth, we assessed symptoms and viral kinetics in a total of 86 patients, including 12 asymptomatic patients, which were not sufficient to draw firm conclusions from. Further research with larger sample sizes, particularly with sufficient numbers of asymptomatic patients, are needed to replicate these findings. Finally, it may be argued that asymptomatic patients might have been diagnosed later than symptomatic patients because of the lack of symptoms, and that this time lag could have affected the viral shedding dynamics. However, considering that most asymptomatic or presymptomatic patients underwent PCR testing for SARS-CoV-2 due to epidemiologic links (data not shown), this bias was not likely to have substantially affected our main findings. 

## 5. Conclusions

In conclusion, compared with symptomatic patients, asymptomatic patients with SARS-CoV-2 infection had a similar viral load during the early course of the disease, but exhibited a more rapid decrease in viral load with the loss of infectivity. Symptomatic patients with COVID-19 had a high infectivity with high symptom scores during the early course of disease and gradually lost infectivity and symptom severity.

## Figures and Tables

**Figure 1 viruses-13-02133-f001:**
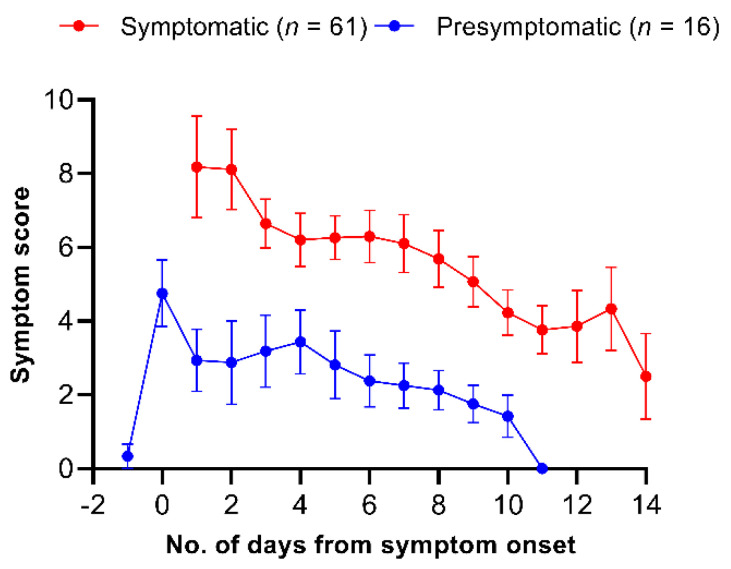
Summary curves of total symptoms scores in symptomatic and presymptomatic patients.

**Figure 2 viruses-13-02133-f002:**
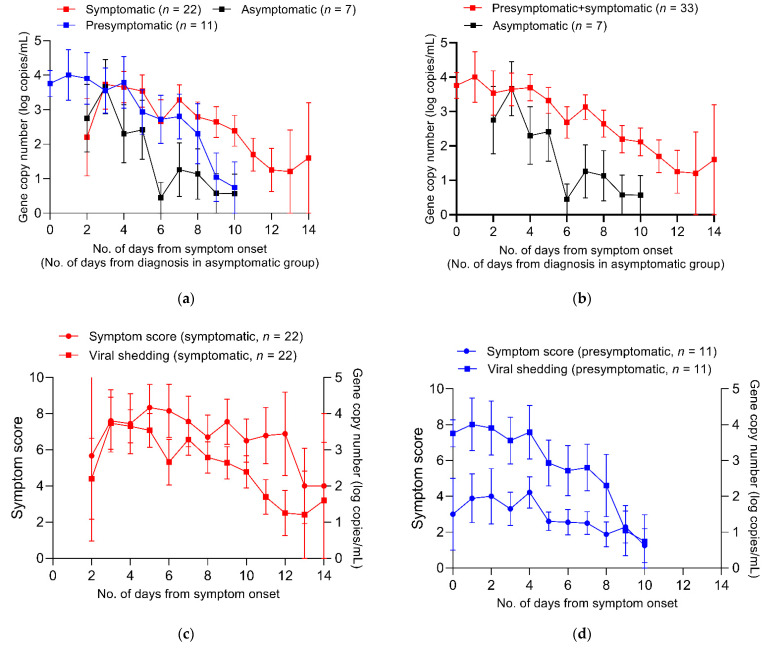
Summary curves of viral shedding and symptom score in patients. (**a**) Summary curves of viral shedding for symptomatic, presymptomatic and asymptomatic patients. (**b**) Summary curves of viral shedding for symptomatic + presymptomatic patients and asymptomatic patients. (**c**) Summary curves of symptoms core and viral shedding for symptomatic patients. (**d**) Summary curves of symptom score and viral shedding for presymptomatic patients.

**Figure 3 viruses-13-02133-f003:**
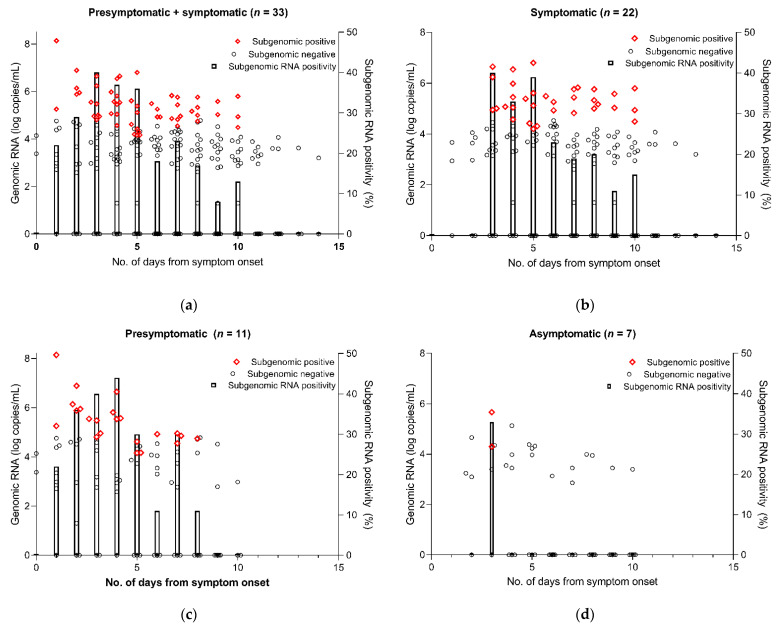
Results of subgenomic RNA test for SARS-CoV-2. The subgenomic positive (red) is plotted as the genomic copy number in a sample (left *y*-axis), where subgenomic RNA was also detected. The subgenomic negative (black) is plotted as the genomic copy number in a sample (left *y*-axis), where no subgenomic RNA was detected. The subgenomic positivity (right *y*-axis) indicates the percentage of subgenomic RNA-positive samples among the total samples by the day. (**a**) Presymptomatic + symptomatic. (**b**) Symptomatic. (**c**) Presymptomatic. (**d**) Asymptomatic.

**Figure 4 viruses-13-02133-f004:**
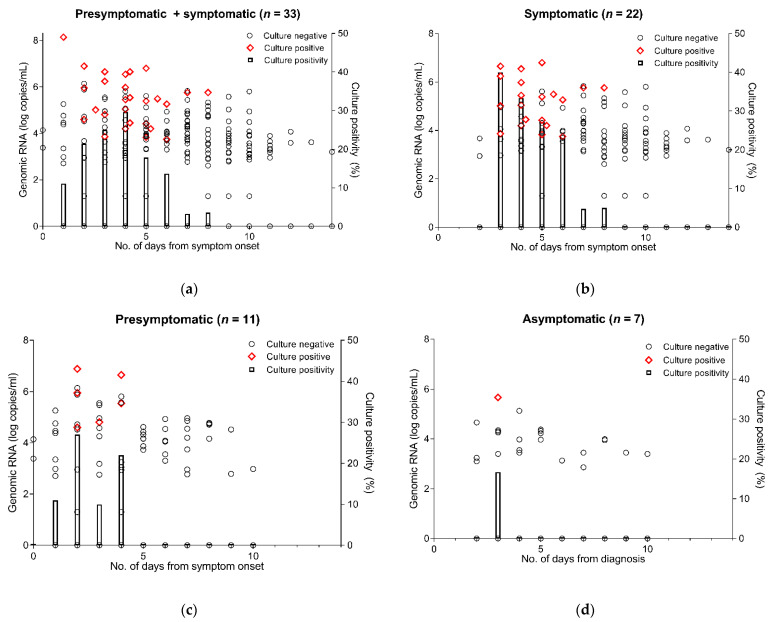
Results of cell culture for SARS-CoV-2. Red-colored dots indicate positive cell culture. The subgenomic positive (red) is plotted as the genomic copy number in a sample (left *y*-axis), where subgenomic RNA was also detected. The subgenomic negative (black) is plotted as the genomic copy number in a sample (left *y*-axis), where no subgenomic RNA was detected. The subgenomic positivity (right *y*-axis) indicates the percentage of subgenomic RNA positive samples among the total samples by the day. (**a**) Presymptomatic + symptomatic. (**b**) Symptomatic. (**c**) Presymptomatic. (**d**) Asymptomatic.

**Table 1 viruses-13-02133-t001:** Baseline characteristics of patients with asymptomatic or mild COVID-19.

Characteristics	Total (*n* = 89)	Asymptomatic (*n* = 12)	Presymptomatic (*n* = 16)	Symptomatic (*n* = 61)	*p* Value
**Sex**					0.35
Female	51 (57.3)	6 (50.0)	7 (43.8)	38 (62.3)	
Male	38 (42.7)	6 (50.0)	9 (56.2)	23 (37.7)	
**Age,** median years (IQR)	49 (34–60)	61 (51–62.5)	58 (42–62)	44 (33–55)	0.001
**Underlying condition**					
Hypertension	20 (22.5)	3 (25.0)	6 (37.5)	11 (18.0)	0.25
Diabetes	8 (9.0)	1 (8.3)	2 (12.5)	5 (8.2)	0.86
Asthma	2 (2.2)	0 (0)	0 (0)	2 (3.3)	0.63
Pregnancy	1 (1.1)	0 (0)	0 (0)	1 (1.6)	0.79
Smoking	12 (13.5)	2 (16.7)	2 (12.5)	8 (13.1)	0.92
**Days from diagnosis to admission**					0.83
0	77 (86.5)	11 (91.7)	14 (87.5)	52 (85.2)	
1	12 (13.5)	1 (8.3)	2 (12.5)	9 (14.8)	
**Days from symptom onset to admission,** median (IQR)	2 (1–3)	NA	0 (0–0)	2 (1–3)	<0.001
**Length of stay,** median days (IQR)	11 (11–11)	11 (11–11)	11 (11–11)	11 (11–11)	0.37
**Clinical course**					0.18
Discharge per protocol	82 (92.1)	12 (100)	16 (100)	54 (88.5)	
Transfer to hospital	7 (7.9)	0 (0)	0 (0)	7 (11.5)	

## Data Availability

Not applicable.

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
