# Peer review of "Dynamics of Viral Shedding and Symptoms in Patients with Asymptomatic or Mild COVID-19"

_viruses, 2021, doi:10.3390/v13112133_

Round 1
Reviewer 1 Report
Bae et al reported a clinical case study to address the question whether there is difference in viral replication at upper respiratory tract between asymptomatic and symptomatic COVID patients. This is an important question because the virus concentration in the upper respiratory tract plays an important role in viral transmission. Their results indicated that 1) in the early phase the viral loads are independent of presence of symptom of COVID; 2) virus waned away faster in asymptomatic patients compared to symptomatic patients. Their finding provides experimental evidence for public health officials to design the policy to better constrain SARS-CoV-2 viruses.
Major Criticism:
- Sample size is too low. There are only total of 89 patients enlisted in this study. Particularly, there are only 12 asymptomatic patients. This diminished the confidence of general applicability of their conclusion.
Author Response
Point 1. Sample size is too low. There are only total of 89 patients enlisted in this study. Particularly, there are only 12 asymptomatic patients. This diminished the confidence of general applicability of their conclusion.
Response 1: We agree to the reviewer’s comment. We described the limitation of small sample size in this study to the Discussion section as below.
In the Discussion section (page 12, lines 346-350).
“…Sixth, we assessed symptoms and viral kinetics in a total 86 patients, including 12 asymptomatic patients, which were not sufficient to draw firm conclusions. Further researches with larger sample sizes, particularly with sufficient numbers of asymptomatic patients, are needed to replicate these findings.”

Reviewer 2 Report
The manuscript entitled Dynamics of viral shedding and symptoms in patients with asymptomatic or mild COVID-19 provides a valuable contribution to our understanding of SARS-CoV-2 infection dynamic that will be of interest to a large number of readers. In general, the manuscript is well written, the experimental approach is appropriate, and the figures and tables are well presented and aid understanding of the data. The data is correctly analysed and interpreted and discussed in relation to previously published works. The limitations of the study are acknowledged.
There are several areas of manuscript which would benefit from the inclusion of a greater level of detail and or clarification as detailed below:
Line 28-29: Is this the correct way around? 10% on day 8, and 5% day 6
Line 61. It is not clear how asymptomatic patients were recruited to the study if the community facilities were for those with mild-COVID. - Did it also include asymptomatic patients who had tested positive for SARS-CoV-2? Please clarify.
Line 90: the word 'until' suggests that patients developed symptoms at 10 days, Do the authors mean up to 10 days after PCR diagnosis? Please clarify.
Line 96. The samples were aliquoted - was there any preservation medium used to help maintain viability / RNA integrity during the freeze thaw process?
Lines 102-104 - please provide the details of procedure including reaction and cycling conditions. These have not been provided in the supplementary data. Please also clarify the limit of detection for this assay and the basis on which a sample is considered positive for genome RNA.
Lines 106-110: Please provide the details of procedure including reaction and cycling conditions. These have not been provided in the supplementary data. Please also clarify the limit of detection for this assay and the basis on which a sample is considered positive for sub genomic RNA and if this is an end point or quantitative assay.
Line 114: Format the number of cells used correctly.
Line 115; Please clarify the volume of the inoculum.
Supplementary Figure 1 should also be referenced appropriately in the materials and methods section.
Figure 3: please clarify in the legend what these graphs are showing. ie: is the sub genomic positivity = the % of genomic RNA positive samples that were also positive for sub-genomic RNA? Is the sub-genomic positive (red) is plotted as the genomic copy number in a sample, where sub-genomic RNA was also detected? Is the sub-genomic negative (black) is plotted as the genomic copy number in a sample, where no sub-genomic RNA was detected?
Line 222: 'rate' please clarify what is meant by 'rate'. - number of positive samples decreasing over time? - The copy number of sub-genomic RNA decreasing over time? What is the limit of detection for the sub genomic analysis?
Line 230 - again please clarify 'rate'
Line 230 - culturs - please correct spelling.
Figure 4: As above comments for figure 3. please clarify the presentation.
Table S1: Please specify the basis on which you would call the samples positive for SARS Cov2 -ie one or both gene targets and the internal control positive?
Table S2 - Are these just primers? I don't see any probe details - please clarify.

Author Response
There are several areas of manuscript which would benefit from the inclusion of a greater level of detail and or clarification as detailed below:
- Line 28-29: Is this the correct way around? 10% on day 8, and 5% day 6
Response: Thank you for the precise comment. In symptomatic or presymptomatic patients, subgenomic RNA was detected up to day 10 with the percentage of subgenomic RNA positivity gradually decreasing over time (from 40% [8/20] at day 3 to 13% [3/24] at day 10), and viable virus by cell culture was detected up to day 8 with the percentage of cell culture positivity gradually decreasing over time (from 40% [8/20] at day 4 to 4% [1/28] at day 8). We have revised the Abstract as follows.
In the Abstract section (page 1, lines 26-32).
From
“…Subgenomic RNA and viable virus by cell culture in asymptomatic patients were detected only until 3 days after diagnosis, the positivity of subgenomic RNA and cell culture in symptomatic patients gradually decreased from both 40% in the early disease course to 10% and 5% at 8 days and at 6 days after diagnosis, respectively. Asymptomatic COVID-19 patients had a similar viral load to symptomatic patients during the early course of disease but exhibited a rapid decrease of viral load with loss of infectivity.”
To
“…Subgenomic RNA and viable virus by cell culture in asymptomatic patients were detected only until 3 days after diagnosis, and the positivity of subgenomic RNA and cell culture in symptomatic patients gradually decreased from both 40% in the early disease course to 10% and 5% at 8 days and at 6 days after diagnosis 13% at 10 days and 4% at 8 days after the symptom onset, respectively. In conclusion, symptomatic patients had high infectivity with high symptom scores during the early disease course and gradually lost infectivity depending on the symptom. Conversely, asymptomatic patients exhibited a rapid decrease of viral load with loss of infectivity, despite a similar viral load during the early disease course. Asymptomatic COVID-19 patients had a similar viral load to symptomatic patients during the early course of disease but exhibited a rapid decrease of viral load with loss of infectivity.”
- Line 61. It is not clear how asymptomatic patients were recruited to the study if the community facilities were for those with mild-COVID. - Did it also include asymptomatic patients who had tested positive for SARS-CoV-2? Please clarify.
Response: We appreciate your thoughtful comments. Asymptomatic patients with positive SARS-CoV-2 PCR as well as mild patients were admitted to community facilities and included in this study. We have clearly described this information in the Materials and Methods section as follows.
In the Materials and Methods section (page 2, lines 60-67).
“In South Korea, adult patients with mild or asymptomatic COVID-19 are isolated in non-hospital community facilities regardless of the accompanying symptoms. All COVID-19 patients admitted to the community facilities were diagnosed by PCR testing for reasons such as COVID-19-related symptoms or contact with other confirmed cases. Between January 10 and February 22, 2021, we recruited adult patients with asymptomatic or mild laboratory-confirmed COVID-19 from a designated non-hospital community facility in Seoul, South Korea, who were willing to record their symptoms in an electronic diary. All the patients were asymptomatic or had only mild symptoms at the time of admission.”
- Line 90: the word 'until' suggests that patients developed symptoms at 10 days, Do the authors mean up to 10 days after PCR diagnosis? Please clarify.
Response: Thank you for your attentive comment. We have corrected the sentence according to the reviewer’s comment.
In the Materials and Methods section (page 3, lines 94-95).
“…iii) asymptomatic patients who did not develop any symptom up to 10 days after PCR diagnosis.”
- Line 96. The samples were aliquoted - was there any preservation medium used to help maintain viability / RNA integrity during the freeze thaw process?
Response: To avoid dilution of the saliva samples, we did not supplement the samples with medium. Since saliva samples were aliquoted and stored for each use, thus freezing-and-thawing was not repeated. The extracted RNA was immediately used for PCR analysis. We added this information in the Materials and Methods section as follows.
In the Materials and Methods section (page 3, lines 101-102).
“Self-collected saliva samples were obtained from patients from the day of study enrollment until the day of discharge. Each day, patients collected a 2-mL volume of saliva into an airtight container provided at admission. Patients were asked to avoid food, water, and teeth brushing for at least 30 minutes prior to sample collection. Saliva samples were picked up within an hour by the medical staff and transported to a designated laboratory where they were aliquoted and stored at –80 °C until use. No preservation or transport medium was used.”
- Lines 102-104 - please provide the details of procedure including reaction and cycling conditions. These have not been provided in the supplementary data. Please also clarify the limit of detection for this assay and the basis on which a sample is considered positive for genome RNA.
Response: Following the reviewer’s comments, we described the detailed procedures of the real-time RT-PCR and the limit of detection for this assay in the Supplemental Materials section as follows.
In the Supplemental Materials (lines 445-453 and lines 497-499)
Measurement of viral load by real-time RT-PCR assay
“Multiplex RT-PCR assay mix (20 μL) contained 4 μL of 5X master mix (LightCycler Multiplex RNA Virus Master, Roche, Basel, Switzerland), 0.1 μL of 200X enzyme mix, 500 nM of each S and N gene primer, 200 nM of each S and N gene probe, 250 nM of internal control primers, 100 nM of internal control probes, and 5 μL of extracted RNA or in vitro-synthesized control RNA. PCR amplification was performed with a LightCycler 96 system (Roche) in the following conditions: reverse transcription at 50℃ for 10 min, initial denaturation at 95℃ for 5 min, 45 cycles of 2-step amplification, denaturation at 95℃ for 10 s, and final extension at 60℃ for 30 s. To generate calibration curves, serial dilutions from 107 to 5 copies/μL of synthetic control RNA were assayed in six independent sets of reactions (Supplementary Figure 4). The detection limit of this assay was 5 copies/reaction (2.6 log copies/ml of specimen) and viral copy numbers were determined by plotting CT values against log copies/reaction.”
Figure S4. Correlation curves for cycle thresholds versus copies of the N and S genes (see attached file)
- Lines 106-110: Please provide the details of procedure including reaction and cycling conditions. These have not been provided in the supplementary data. Please also clarify the limit of detection for this assay and the basis on which a sample is considered positive for sub genomic RNA and if this is an end point or quantitative assay.
Response: As reviewer’s comments, we have added detailed procedures of real-time RT-PCR for detecting subgenomic RNA and a cut-off value for positive result as follows.
In the Supplemental Materials (lines 455-458)
Detection of N and S gene subgenomic RNAs
“Multiplex real-time RT-PCR assay reaction mixture and cycling condition were same as the condition of the RT-PCR for viral loads measurement described in the Supplemental Materials as above. The result was considered as a positive if the cycle threshold values of both N and S subgenomic RNAs were less than 36. The limit of detection of this assay was 5 copies/reaction.”
- Line 114: Format the number of cells used correctly.
Response: We have corrected the number of cells as follows (page 3, line 120).
From
“Vero cells were cultured in 6-well plates with 9 ï‚´ 105 cells/well about 24 h earlier.”
To
“Vero cells were cultured in 6-well plates with 9 × 105 cells/well about 24 h earlier.”
- Line 115: Please clarify the volume of the inoculum.
Response: We thank for the reviewer’s valuable advice. Following the reviewer’s suggestion, we have clarified the volume of the inoculum as follows.
In the Materials and Methods section (page 3, line 121).
“200 ul of each diluted sample was inoculated into cells and incubated for 1 h (37℃, 5% CO2) with rocking every 15 min, and overlaid with 2 mL of Dulbecco's Modified Eagle Medium/Nutrient Mixture F12 (DMEM/F-12) medium containing 0.6% oxoid agar.”
- Supplementary Figure 1 should also be referenced appropriately in the materials and methods section.
Response: According to the reviewer’s suggestion, we added citation of the Supplementary Figure 1 in the Materials and Methods section as follows.
In the Materials and Methods section (page 2, lines 67-69).
“Patients who agreed with daily saliva sample collection were enrolled for the viral kinetic study (Supplementary Figure 1). Those who were younger than 18 years of age or who were pregnant were excluded. All admitted patients were monitored for symptoms related to COVID-19.”
- Figure 3: please clarify in the legend what these graphs are showing. ie: is the sub genomic positivity = the % of genomic RNA positive samples that were also positive for sub-genomic RNA? Is the sub-genomic positive (red) is plotted as the genomic copy number in a sample, where sub-genomic RNA was also detected? Is the sub-genomic negative (black) is plotted as the genomic copy number in a sample, where no sub-genomic RNA was detected?
Response: The left y-axis represents the genomic RNA copy number of corresponding samples, and red-colored dots and black-colored dots indicate positive and negative subgenomic RNA, respectively. So, the subgenomic positive (red) is plotted as the genomic copy number in a sample, where subgenomic RNA was also detected. The subgenomic negative (black) is plotted as the genomic copy number in a sample, where no subgenomic RNA was detected. The right y-axis represents the proportion of subgenomic RNA-positive samples out of the total samples by the day. So, the subgenomic positivity indicates the percentage of genomic and subgenomic RNA positive samples among the total samples by the day. We have described this information more clearly in the figure legend as follows (page 9, lines 218-222).
“Figure 3. Results of subgenomic RNA test for SARS-CoV-2. Red-colored dots indicate positive subgenomic RNA. Black colored dots indicate negative subgenomic RNA. Bars indicate the positivity of subgenomic RNA among the tested specimens. The subgenomic positive (red) is plotted as the genomic copy number in a sample (left y-axis), where subgenomic RNA was also detected. The subgenomic negative (black) is plotted as the genomic copy number in a sample (left y-axis), where no subgenomic RNA was detected. The subgenomic positivity (right y-axis) indicates the percentage of subgenomic RNA positive samples among the total samples by the day. A. presymptomatic + symptomatic. B. symptomatic. C. presymptomatic. D. asymptomatic.”
- Line 222: 'rate' please clarify what is meant by 'rate'. - number of positive samples decreasing over time? - The copy number of sub-genomic RNA decreasing over time? What is the limit of detection for the sub genomic analysis?
Response: Reviewer’s point was well-taken. We tried to explain that the proportion of the positive subgenomic RNA among the tested samples was decreasing over time. The lower limit of detection (LoD) of the subgenomic RNA was 5 copies/mL, and samples with both S gene and N gene positive in subgenomic RNA testing was considered as positive. We changed the expression of the “rate” to the “percentage” as follows (page 9, line 230).
“In symptomatic and presymptomatic patients, subgenomic RNA was detected up to day 10 and day 8 after symptom onset, respectively, with the rate percentage of specimens with positive subgenomic RNA gradually decreasing over time.”
- Line 230 - again please clarify 'rate'
Response: We changed the expression of the “rate” to the “percentage” as follows (page 10, line 238).
“In symptomatic and presymptomatic patients, cell culture revealed positive results up to day 8 and day 4 after symptom onset, respectively, with the rate percentage of cell culture positivity gradually decreasing over time.”
- Line 230 - culturs - please correct spelling.
Response: We have revised this typo-error in the revised text as follows (page 10, line 239).
From
“culturs”
To
“cultures”
- Figure 4: As above comments for figure 3. please clarify the presentation.
Response: We revised the figure legend of Figure 4 as follows (page 10, lines 243-248).
Figure 4. Results of cell culture for SARS-CoV-2. Red-colored dots indicate positive cell culture. Black colored dots indicate negative cell culture. Bars indicate the positivity of cell culture among the tested specimens. The subgenomic positive (red) is plotted as the genomic copy number in a sample (left y-axis), where subgenomic RNA was also detected. The subgenomic negative (black) is plotted as the genomic copy number in a sample (left y-axis), where no subgenomic RNA was detected. The subgenomic positivity (right y-axis) indicates the percentage of subgenomic RNA positive samples among the total samples by the day. A. presymptomatic + symptomatic. B. symptomatic. C. presymptomatic. D. asymptomatic.
- Table S1: Please specify the basis on which you would call the samples positive for SARS Cov2 -ie one or both gene targets and the internal control positive?
Response: The saliva samples were considered positive for SARS-CoV-2 genomic or subgenomic RNA detection if both N and S genes were positive as well as the positive internal control. We added the footnote in the Table S1 as follows (lines 462-463).
Table S1. Primers and probes used for real-time RT-PCR assay for the detection of N gene and S gene
|
Target (Accession #) |
Name |
Location |
Sequence |
Modification |
|
N gene (NC_045512) |
NF |
29356 |
AACATTCCCACCAACAGAGC |
|
|
NR |
29529 |
GCCTGAGTTGAGTCAGCACT |
|
|
|
NP |
29462 |
GCTGATGAAACTCAAGCCTTACCGCA |
5’Cy5, 3’BHQ2 |
|
|
S gene (NC_045512) |
SF |
21624 |
GAACTCAATTACCCCCTGCAT |
|
|
SR |
21787 |
ACCATTGGTCCCAGAGACAT |
|
|
|
SP |
21657 |
TCACACGTGGTGTTTATTACCCTGACA |
5’FAM, 3’BHQ1 |
|
|
Internal control (NC_000007.14) |
BAF |
1670 |
ACTAACACTGGCTCGTGTGA |
|
|
BAR |
1774 |
CTTGGGATGGGGAGTCTGTT |
|
|
|
BAP |
1700 |
AGGCTGGTGTAAAGCGGCCTTGG |
5’HEX, 3’BHQ1 |
Footnote. The samples were considered positive for SARS-CoV-2 genomic RNA detection if both N and S genes were positive as well as the positive internal control.
- Table S2 - Are these just primers? I don't see any probe details - please clarify.
Response: We are sorry. The corrected primers and probes have been replaced in Supplementary Table 2 as follows (lines 465-467).
Table S2. Primers and probes used for the detection of N gene and S gene of subgenomic RNAs of SARS-CoV-2
|
Target* |
Name |
Location |
Sequence |
Modification |
|
5’ leader |
SG-F |
15 |
CCTTCCCAGGTAACAAACCA |
|
|
N gene |
SGRT-NR2 |
28436 |
CGGTGAACCAAGACGCAGTA |
|
|
|
SGRT-NP2 |
28322 |
TTTGGTGGACCCTCAGATTCAACTGG |
5’-FAM, 3’-BHQ1 |
|
S gene |
SGRT-SR1 |
21678 |
GGGTAATAAACACCACGTGTGAA |
|
|
|
SGRT-SP1 |
21617 |
ACAACCAGAACTCAATTACCCCCTGCA |
5’-CY5, 3’-BHQ2 |
(*Accession #: NC_045512.2)

Round 2
Reviewer 1 Report
The authors have modified the manuscript to address my main concern.